# Crosstalk between Selenium and Sulfur Is Associated with Changes in Primary Metabolism in Lettuce Plants Grown under Se and S Enrichment

**DOI:** 10.3390/plants11070927

**Published:** 2022-03-30

**Authors:** Muna Ali Abdalla, Christine Lentz, Karl H. Mühling

**Affiliations:** Institute of Plant Nutrition and Soil Science, Faculty of Agricultural and Nutritional Sciences, Kiel University, Hermann-Rodewald-Str. 2, 24118 Kiel, Germany; stu201981@mail.uni-kiel.de

**Keywords:** Se-and-S interaction, foliar application, mineral elements, amino acids, soluble sugars, organic acids, primary metabolites

## Abstract

This study investigated the beneficial effects of selenium (Se) and sulfur (S) enrichment on the primary metabolism in butterhead lettuce. The plants were treated with three levels of Se via foliar application in the presence of two S levels in the nutrient solution under greenhouse conditions. The lettuce plants that were exposed to the lower selenate level (1.3 μM) in combination with the adequate and high S supplies (1 and 2 mM, respectively) accumulated 38.25 ± 0.38 µg Se g^−1^ DM and 47.98 ± 0.68 µg Se g^−1^ DM, respectively. However, a dramatic increase in the Se concentration (122.38 ± 5.07 µg Se g^−1^ DM, and 146.71 ± 5.43 µg Se g^−1^ DM, respectively) was observed in the lettuce heads that were exposed to the higher selenate foliar application (3.8 μM) in response to the varied sulfate concentrations (S1 and S2, respectively). Under higher Se and S supplies in the lettuce plants, the levels of organic acids, including malic acid and citric acid, decreased therein to 25.7 ± 0.5 and 3.9 ± 0.3 mg g^−1^ DM, respectively, whereas, in the plants that were subjected to adequate S and lower Se fertilization, the malic acid, and citric acid levels significantly increased to 47.3 ± 0.4 and 11.8 ± 0.4 mg g^−1^ DM, respectively. The two Se levels (1.3 and 3.8 μM) under the S1 conditions also showed higher concentrations of water-soluble sugars, including glucose and fructose (70.8.4 ± 1.1 and 115.0 ± 2.1 mg g^−1^ DM; and 109.4 ± 2.1 and 161.1 ± 1.0 mg g^−1^ DM, respectively), compared to the control. As with the glucose and fructose, the amino acids (Asn, Glu, and Gln) exhibited strikingly higher levels (48.7 ± 1.1 μmol g^−1^ DM) under higher S and Se conditions. The results presented in this report reveal that the “crosstalk” between Se and S exhibited a unique synergistic effect on the responses to the amino acids and the soluble sugar biosynthesis under Se and S enrichment. Additionally, the Se-and-S crosstalk could have an important implication on the final nutritional value and quality of lettuce plants.

## 1. Introduction

Selenium (Se) is considered to be a useful element for plants because it has remarkable effects on antioxidants and, thus, it promotes plant productivity and resistance to oxidative stress [1]. For humans and animals, it is known to be an essential micronutrient for the proper functioning of the immune system and many organs, including the heart, thyroid, brain, prostate, and testis [2]. Furthermore, Se has demonstrated anticancer bioactivity due to its significant role in the modulation of intracellular redox-regulated transcription factors [3]. Consequently, it can act both as an antioxidant and as a pro-oxidant, depending on certain concentrations in the organism [4]. An inadequate dietary intake of Se can cause health complications, including heart disease, infertility, cognitive decline, and myodegenerative diseases [5]. In ruminant animals, Se deficiency has been implicated in white muscle disease, which is a condition that is characterized by muscle weakness, heart failure, and livestock death [3]. Studies have shown that the use of Se enrichment has been successful in raising the Se levels in food crops [6,7].

Sulfur (S) is one of the important nutrients for the growth and metabolism of plants. S is a highly reactive element, which has a multitude of different oxidation states [8]. It is present in plants in several forms, and it is involved in many chemical reactions. S is a component of proteins; the amino acids, cysteine (Cys) and methionine (Met), vitamins, cofactors, and a plethora of primary and secondary metabolites [9]. In addition, S is known as a healing mineral because of the central role of S-containing compounds (i.e., glucosinolates and phytoalexins from Brassica crops, and allicin and other organosulfur metabolites from garlic) in improving food quality, as well as in the disease tolerance and resistance of plants, besides their use as herbal medicinal products [10].

Because of the similar chemical and physical properties of Se and S, S can be substituted with Se in the metabolism of plants. This leads to their competition in the uptake, transport, and assimilation in plants. Selenate (SeO_4_^2−^) is reduced via the same assimilation pathway as sulfate (SO_4_^2−^), and it is incorporated into Se-containing amino acids, such as selenocysteine (SeCys) and selenomethionine (SeMet) [11]. This process replaces the S-containing amino acids, cysteine, and methionine. It is assumed that SeCys and SeMet can be incorporated into proteins. However, the reduced Se cannot replace most of the functions of the sulfide, such as the Fe–S clusters, the disulfide bridges, and the catalytic processes. As a result, the stability and the functional activity of the proteins are impaired; and at high Se concentrations, Se toxicity occurs in the plants that do not accumulate Se [12]. However, some plants have developed strategies to prevent Se toxicity, such as Se hyperaccumulator plants [13]. Lettuce plants have been reported to have a good capacity to tolerate Se and to have enhanced antioxidant properties [14]. It is noteworthy that primary metabolites, such as carbohydrates, organic and amino acids, and vitamins, in addition to the secondary metabolites, are necessary for plant growth and metabolic processes, stress acclimation, and defense against herbivores and pathogens. Furthermore, these metabolites can also define the nutritional quality of fruits and vegetables, the color, taste, smell, antioxidant potential, and other pharmacological properties [15]. Se intakes across Europe have been reported to be lower than in the United States, which is due to insufficient soil levels, particularly in Eastern Europe [16]. Additionally, farmers often do not have preferences for the application of Se to their soils, which is due to its nonessentiality for the plant’s growth. Subsequently, this makes S available so that the plant’s S transporters show low affinity and prefer to uptake S rather than Se because of their antagonistic interaction. Therefore, the current report recommends Se foliar application as a novel fertilizer tool in the presence of S, supplied via root application. Accordingly, the enrichment of Se by this method ensures the availability of Se in plants. Thus, the consumption of Se-enriched lettuce may boost human health. Paradoxically, however, little is known about the impact of the sulfate-and-selenate interaction on the primary metabolism in lettuce plants that are grown under S and Se enrichment.

This study reports on the influence of the Se and S supply on the biosynthesis of amino acids, water-soluble sugars, and organic acids in butterhead lettuce. We tested the hypothesis that the Se-and-S crosstalk will induce the biosynthesis of primary metabolites, especially amino acids, and soluble sugars. Additionally, the unique synergistic effect between Se and S due to Se foliar application will improve the nutritional quality of the lettuce.

## 2. Results

### 2.1. Plant Biomass

The results for the plant growth are presented using the yield and the DM accumulation. Additionally, the lettuce heads were visually compared shortly after the harvest in order to record the influence of the different treatments on the plants (Figure 1).

Figure 1 shows the side views of the lettuce heads that were grown in a hydroponic system and treated with three Se foliar applications under two S levels. All the lettuce heads showed a normal green color, which is a characteristic of healthy plants, with only slight differences in appearance and size.

Concerning the yield and DM accumulation, the results indicate some significant differences between the varied treatments, as shown in Figure 2. Under the S1 and Se1 conditions, the lettuce plants’ yield and total-plant DM did not exhibit significant changes. On the contrary, the increased Se concentration (Se2) resulted in a significantly greater reduction in the yield and DM than in the control (Se0/S1) (Figure 2A,B). On the other hand, the lettuce heads treated with Se1 and Se2 did not show any significant difference in the yield and DM under higher S conditions (S2) compared to the control (Se0/S2).

### 2.2. Se Accumulation in Lettuce Plants

Se biofortification in food crops is a valuable strategy for mitigating the nutritional Se deficiency in humans [7]. Moreover, the availability of S has a major impact on the Se accumulation because of the effects of the competition of the two oxyanions. Additionally, Se influences S uptake by interfering with the intrinsic regulatory mechanisms. The total Se concentrations were determined to evaluate the effects of different Se levels (Se0: 0; Se1: 1.3; and Se3: 3.8 µM) under varied S conditions (S1: 1; and S2: 2 mM) on the Se accumulation in lettuce plants. The results indicate that two varied Se foliar applications (Se1 and Se2) caused significant increases in the Se concentrations in the lettuce heads, which was not the case with the zero variant (Se0) (Figure 3). The lettuce plants that were exposed to the Se1 treatment accumulated 38.25 ± 0.38 µg Se g^−1^ DM and 47.98 ± 0.68 µg Se g^−1^ DM under the S1 and S2 conditions, respectively. However, a dramatic increase in the Se accumulation was observed in the lettuce plants that were exposed to the elevated selenate treatment (Se2) under both S supplies. High levels of Se (122.38 ± 5.07 µg Se g^−1^ DM; and 146.71 ± 5.43 µg Se g^−1^ DM, respectively) were accumulated under the S1 and S2 conditions (Figure 3).

### 2.3. Elemental Composition

The uptake of various minerals (macro and micro) that have particular functions in the human body can be either positively or negatively affected by Se biofortification. The mineral profiles in the lettuce plants are shown in Table 1. The Se foliar application (Se1) resulted in significant increases in the P levels in the lettuce heads under adequate and higher S treatments (by 8 and 7% under S1 and S2, respectively). Conversely, Se treatments with either the lower dose (Se1) or the higher dose (Se2) under both S conditions did not significantly alter the Mg^2+^ content. The elevated Se level (Se2) under the adequate S condition (S1) resulted in a significant decrease (8%) in the K^+^ level in the lettuce heads. The Ca^2+^ concentration increased by 9% in response to the Se1 application under the S1 condition, however, it remained unaltered in the lettuce heads in response to both Se levels (Se1 and Se2) under the S2 condition, compared to Se0/S2. Moreover, the N level remained unaffected by the low and high Se concentrations (Se1 and Se2, respectively) under the S1 and S2 conditions in comparison to the Se0/S1 and Se0/S2 controls, respectively. Concerning the total S concentration, an antagonistic response was observed, where the S level decreased by 4% under the Se2/S1 conditions, but a synergistic effect was detected under the Se2/S2 conditions, where the S increased by 14% in comparison to the Se0/S2 control. With regard to the accumulation of micronutrients, the results show a significant increase in the Mn^2+^, Cu^2+^, and Zn^2+^ levels (by 22, 9, and 7%, respectively) under the Se1/S1 conditions. However, the Fe concentration remained unchanged under the same conditions. Dramatic decreases in the Zn^2+^ concentration (by 42 and 39%, respectively) were observed under the Se2/S1 and Se2/S2 treatments, respectively. Furthermore, the Mn^2+^ level significantly decreased by 42% under higher S and Se-deficient conditions. However, it was unchanged under both the Se1 and Se2 applications, whereas the Fe and Cu^2+^ concentrations increased by 13 and 4%, respectively, under the Se1/S2 supply, and declined again by 7 and 41%, respectively, in response to the higher Se treatment (Se2)

### 2.4. Amino Acids Analysis

To examine the effect of the varied Se and S treatments on the nutritional quality, the free individual amino acid levels in the lettuce heads were determined. The current results indicate that the amino acids in the lettuce heads included: aspartic acid (Asp); threonine (Thr); serine (Ser); asparagine; glutamic acid and glutamines (Asn, Glu, Gln); glycine (Gly); alanine (Ala); valine (Val); isoleucine (Ile); and leucine (Leu). In particular, Asn, Glu, and Gln showed overlapping peaks, and their concentrations were added up as the sum of the accumulation of the three amino acids. The Se treatment (Se1) did not dramatically affect the accumulation of the amino acids (including Thr, Ser, Asn, Glu, Gln, Gly, Val, Ile, and Leu) under an adequate S level. However, the Asp level was significantly enhanced (by 22%) under the same conditions, while the Ala concentration decreased (by 11%) in response to the previous treatment (Table 2). A significant decrease was observed in the Thr, Ser, Asn, Glu, Gln, Val, and Ile levels (by 18, 13, 5, 14, and 14%, respectively) under the higher Se application (Se2) and adequate S treatment, in comparison to the control (Se0/S1). Se and S demonstrated a synergistic relationship with regard to Asp and Asn, Glu, and Gln accumulation, where their levels increased under the Se1/S2 (by 14 and 24%, respectively) and Se2/S2 (by 26 and 37%, respectively) treatments, compared to the control (Se0/S2). The concentrations of Val, Ile, and Leu were significantly induced (by 20, 34, and 70%, respectively) under the Se1/S2 application.

### 2.5. Determination of Organic Acids, Water-Soluble Sugars, and Inorganic Anions

The results show significant variations in the water-soluble low-molecular-weight organic acids and sugars, in addition to the inorganic anions. These organic and inorganic constituents can substantially contribute to improving crop quality, with associated implications for human nutrition and health.

A significant decrease in the malic acid under the higher Se level (Se2) and the lower S treatment (S1) was observed. However, the malic acid level significantly increased under the lower Se level (Se1) and the higher S treatment (S2) (Figure 4A), while the oxalic acid and citric acid levels decreased under the lower Se (Se1) and the higher S treatment (S2) (Figure 4B,C). As for the water-soluble sugars, the glucose, fructose, and sucrose levels significantly increased in response to the Se1 treatment under the S1 condition (Figure 4D–F). A dramatic accumulation of glucose and fructose was observed under the higher Se condition (Se2/S1), while the elevated S level (S2) demonstrated a dramatic decrease under the Se-limited condition (Figure 4D,E). The sucrose level remained unaltered among the lettuce plants in response to the Se1 treatments under the S2 condition.

Apart from the water-soluble sugars, the levels of various inorganic anions (i.e., Cl^−^, SO_4_^2−^, NO_3_^−^, and PO_4_^3−^) were precisely determined under the varied Se treatments and the two S conditions (Table 3). The Cl^−^ concentration was significantly diminished under the higher Se treatment (Se2) and adequate S (S1), in addition to the lower Se (Se1) and higher S conditions (by 16 and 22%, respectively). Additionally, the SO_4_^2−^ decreased significantly in response to Se1 and Se2 under the S1 conditions (by 27 and 37%, respectively), while the SO_4_^2−^ concentration remained unaffected under the higher S (S2) and the two varied Se conditions.

The NO_3_^−^ concentrations significantly decreased in the lettuce heads under the adequate S treatment and the two Se applications (Se1 and Se2) (by 24 and 40%, respectively) in comparison to the control (Se0/S1), whereas significant decreases in the NO_3_^−^ and PO_4_^3−^ levels (by approximately 26 and 28%, respectively) were observed in response to the higher S and low or high Se fertilization (Table 3).

## 3. Discussion

Lettuce (*Lactuca sativa* L., Asteraceae) is a widely consumed leafy vegetable in the world. Butterhead lettuce is also known for its remarkable nutritive value, including its high vitamin C content. Additionally, lettuce leaves are higher in iron, folate, and potassium than iceberg or multi-leaf lettuce variants [17]. Previous studies have revealed the great capacity of lettuce to accommodate Se in its edible parts, which have demonstrated the effectiveness of Se biofortification in food crops [7]. In this study, different Se and S levels were applied to lettuce heads to investigate their influence on the nutritional quality and the primary metabolites of butterhead lettuce plants, including their amino acids, soluble sugars, and organic acids.

With regard to the plant biomass, the results of this study show that the Se foliar application (Se1: 1.3 µM) in the presence of 1 mM of S supply (S1) in the nutrient solution did not affect the plant growth, including its yield and total-plant DM (Figure 2A,B). Additionally, its yield and DM remained unchanged under the Se doses of 1.3 and 3.8 µM and the higher S conditions (S2), in comparison to the control (Se0/S2).

The Se concentrations were determined in the butterhead lettuce to evaluate the extent of the effects of the Se-and-S interaction on the Se accumulation. The results reveal that the Se level could be associated with changes in the Se and S treatments.

The Se foliar application enhanced the Se accumulation in the butterhead lettuce plants enough to contribute to the Se daily recommended dietary allowance of 70 μg day^−1^ for adult men [18]. Considering that lettuce contains about 95% water [19], our findings indicate that consuming 100 g of fresh lettuce heads treated with Se1 (1.3 µM) under sufficient and higher S (1 mM and 2 mM) conditions can contribute to Se dietary requirements with 191 and 240 μg Se day^−1^, respectively [20,21]. Accordingly, we recommend a daily intake of up to 37 and 29 g (fresh weight) of butterhead lettuce heads grown under Se1/S1 and Se1/S2 conditions, respectively, for adult men (70 μg Se day^−1^) to cover their daily dietary allowance. Furthermore, the lettuce plants accommodated high Se in response to elevated Se treatment (3.8 µM) under both the S1 and S2 conditions (Figure 3). Tian et al. (2017) reported that the SO_4_ supply is critical for the Se acquisition and toxicity. Therefore, Se can be highly toxic to plants under S-limiting conditions [22]. Although increasing the concentration of antagonistic elements such as S could decrease the Se accumulation in plant tissues, in this study, the lettuce plants accumulated higher Se (146.71 ± 5.43 µg Se g^−1^ DM) under a high S supply (Se2/S2), compared to adequate S treatment (Se2/S1), where the Se level was 122.38 ± 5.07 µg Se g^−1^ DM under the same Se level (Se2). This can be attributed to the synergistic relationship between Se and S that is due to the Se foliar application. Furthermore, 100 g of fresh-weight lettuce heads treated with a high Se concentration under S1 and S2 conditions can deliver 612 μg Se day^−1^ and 734 μg Se day^−1^ of the Se requirements, respectively. In this regard, our study recommends that 11 and 10 g of fresh-weight lettuce grown under Se2/S1 and Se2/S2 conditions, respectively, can be consumed daily to cover the daily Se allowance for adult men. The European Food Safety Authority (EFSA) recommends a maximum Se intake for adults of 300 µg Se day^−1^ to avoid symptoms of poisoning [23]. Additionally, a previous study on the relationship between the Se intake and the occurrence of symptoms of selenosis and further biochemical changes in the blood and urine among adults in China indicated that a Se level of 750–850 μg Se day^−1^ may be the marginal concentration of safe intake. The authors also suggested that, because of other variable factors, 400 μg Se day^−1^ could be the maximum safe daily dose [24].

Overall, the results of the elemental composition of the lettuce heads show that the Se and S enrichment affected other minerals that are relevant to human health. Ca^2+^, Mg^2+^, P, K^+^, Mn^2+^, Fe^2+^, Cu^2+,^ and Zn^2+^ are reported to be key nutrients in butterhead lettuce [17]. The application of Se (Se1) under adequate S supply (S1) significantly increased the P and Ca^2+^ levels in the lettuce heads. Additionally, in our recent study, the Se foliar application enhanced the P, K^+^, and Ca^2+^ accumulation in green multi-leaf lettuce under different Se foliar applications [7].

Concerning the micronutrients levels under Se and S enrichment, significant increases in the accumulation of Mn^2+^, Cu^2+^, and Zn^2+^ were observed in response to the Se1/S1 treatment, whereas the Zn^2+^ level declined under the Se2/S1 and Se2/S2 supplies, respectively. Additionally, the Mn^2+^ level decreased in response to the higher S and the Se-limiting conditions. The decline in the Zn^2+^ and Mn^2+^ concentrations under the higher S treatment may be attributed to the antagonistic interaction of S with these minerals. This is indicated in a previous study on the effect of N and S on the nutrient usage and accumulation in onions, in which S showed antagonism towards B, Fe^2+^, Mn^2+^, and Zn^2+^ [25]. In contrast, in this study, the Fe^2+^ and Cu^2+^ accumulated under the Se1S2 treatment and decreased under the Se2 treatment, which shows the antagonistic effect of Se towards Fe^2+^ and Cu^2+^.

Apart from the mineral elements, the results demonstrate that Se-and-S crosstalk can affect the amino acids profile. Amino acids are known as important nutrients because of their functions in the primary and secondary metabolisms in plants. Moreover, free amino acids play a critical role in the taste of vegetable crops, where high levels of Glu, Asp, Ser, Val, Ala, Pro, and Gln were detected in vegetables that included tomato, cucumber, carrot, and pumpkin [26]. The current findings indicate that the butterhead lettuce showed some variations in its free amino acid levels. Consequently, the Se-and-S interaction significantly modified the amino acid levels negatively or positively and, in some cases, did not affect their accumulation, as is shown in Table 2. Asp accumulated when the lettuce plants were treated with Se1/S1 (1.3 μM/1 mM), Se1/S2 (1.3 μM/2 mM), and Se2/S2 (3.8 μM/2 mM), and the Asn, Glu, and Gln total concentrations increased under the Se1/S2 and Se2/S2 treatments. Ramos et al. (2011) reported that the most predominant amino acids in lettuce plants are glutamic acid, arginine, aspartic acid, serine, and threonine [14]. Asp acid has nutritional significance in plants because of its important role in the Asp family pathway, which leads to the biosynthesis of the essential amino acids, including Lys, Thr, Met, and Ile. These amino acids cannot be made by humans and livestock and, as a result, they must be obtained from food [27]. Additionally, the Glu acid family includes the amino acids that are synthesized from glutamate: Gln, Pro, and Arg [28].

The S-containing amino acids, methionine (Met) and cysteine (Cys) were, unfortunately, not investigated in this study because of their rapid oxidation and instability. Several studies, however, indicated that the changes in the S assimilatory pathways that are induced by Se enrichment intervene with N metabolism, and, consequently, affect amino acids and protein biosynthesis [29]. In a recent study on the influence of varied Se applications (0–40 µM) on S metabolism and the accumulation of metabolites in two rocket species (*Eruca sativa* Mill. and *Diplotaxis tenuifolia*) that were grown in a hydroponic system, the authors indicated that all of the Se treatments in the *E. sativa* dramatically reduced the Met level, in comparison to the untreated plants, while, in *D. tenuifolia*, the Cys concentration decreased only after the 40 µM Se treatment, and the Met concentration was unaltered [30]. Moreover, the recent study by Piñero et al. (2022) reported that a Se concentration of 4 or 8 µmol L^−1^ can accumulate all amino acids. Thus, an increase in amino acids could improve the taste of the lettuce and could also provide health benefits for the consumer [31]. In connection with the present results, it can be confirmed that the enrichment of Se and S has a predominantly positive influence on the biosynthesis of amino acids.

With regard to organic acids, our results show a decline in the malic acid level under the Se2/S1 treatment. Additionally, the malic acid and citric acid concentrations decreased in response to the higher S treatment (S2) under the Se-limiting conditions, while the malic acid level increased under the Se1/S2 treatment (Figure 4A). This can be attributed to the role that Se plays in increasing the malic acid concentration. Hu et al. (2022) studied, individually and in combination, silicon (Si) and Se foliar application in cucumber under field conditions and found that the Se application increased the total organic acid level by increasing the concentrations of citric acid, malic acid, and oxalic acid [32]. Organic acids and water-soluble sugars are important metabolites that account for the taste and affect other organoleptic properties of fruits and vegetables, including their color, flavor, and aroma [33]. Our results demonstrate that Se and S showed a synergistic interaction with regard to the glucose, fructose, and sucrose levels, which accumulated under the Se1/S1 treatment in comparison to the control (Se0/S1). The glucose and fructose concentrations were enhanced significantly under the Se2/S1 condition (Figure 4D,E), and both sugars declined significantly under the S2- and Se-limiting conditions. Glucose is highly associated with the perception of sweetness [34].

The Cl^−^ level decreased under the Se2/S1 and Se1/S2 treatments. An adequate Cl^−^ level has the health benefits of the maintenance of the acid and electrolyte balance; the transport of water, minerals, and gases; and the improvement of kidney and muscle function [35]. The SO_4_^2−^ concentration was antagonized by the low and high Se levels (Se1 and Se2) under the S1 treatment, but it was unaltered under the higher S2 and the two varied Se conditions (Table 3). The NO_3_^−^ level decreased significantly under the Se1/S1, Se2/S1, Se1/S2, and Se2/S2 treatments, whereas the PO_4_^3−^ concentration decreased under Se1/S2 and Se2/S2 (Table 3). The tendency of the NO_3_^−^ level to decrease in response to Se biofortification and/or S treatment has also been shown in many previous studies [36].

Hence, leafy vegetables, such as raw spinach, beets, celery, and lettuce, have high NO_3_^−^ accumulation capacities, as they are known as the primary sources of NO_3_^−^. Additionally, processed meats are other sources of nitrite (NO_2_^−^), which can increase our dietary consumption of NO_3_^−^. In the meat industry, NO_3_^−^ and NO_2_^−^ are used as additives [37]. Therefore, an excessive NO_3_^−^ level triggers the oxidation of hemoglobin, which may cause methemoglobinemia in young children. It may also lead to the endogenous accumulation of carcinogenic N-nitroso molecules in adults [38]. A high serum PO_4_^3−^ concentration may lead to cardiovascular diseases [39]. Accordingly, the synergistic interaction between Se and S, which reduced the NO_3_^−^ and PO_4_^3−^ levels, has a tremendous influence on human health.

## 4. Materials and Methods

### 4.1. Plant Material and Treatments

The green butterhead lettuce cultivar named “Pazmanea RZ” was being hydroponically cultivated under standard greenhouse conditions (18 °C day/14 °C night cycle, and a 14 h photoperiod). The seedlings were planted individually in 10 L black containers. The basal nutrient solution was made up of the following nutrients: Macronutrients (mM) Ca(NO_3_)_2_ = 2; NH_4_H_2_PO_4_ = 0.5; MgCl_2_ = 0.5; KNO_3_ = 2; and micronutrients (μM) Fe-EDTA = 200; H_3_BO_3_ = 10; MnSO_4_ = 2; ZnSO_4_ = 0.5; CuSO_4_ = 0.3; and (NH_4_)_2_Mo_7_O_24_ = 0.01. The nutrient solution was changed weekly. There were four replicates that were arranged in a completely randomized design. To investigate the influence of an increasing Se supply on the biosynthesis of the primary metabolites of lettuce plants under different S treatments, we included the foliar application of Se at three levels (Se0, Se1, and Se2) under two S levels (S1 and S2) in the nutrition solution. The tested treatments were as follows: (1) Control (Se0S1: 0 µM Na_2_SeO_4_ + 1 mM K_2_SO_4_; and Se0S2: 0 µM Na_2_SeO_4_ + 2 mM K_2_SO_4_); (2) Se1 + S1 (1.3 µM Na_2_SeO_4_ + 1 mM K_2_SO_4_); (3) Se1 + S2 (1.3 µM Na_2_SeO_4_ + 2 mM K_2_SO_4_); (4) Se2 + S1 (3.8 µM Na_2_SeO_4_ + 1 mM K_2_SO_4_); and (5) Se2 + S2 (3.8 µM Na_2_SeO_4_ + 2 mM K_2_SO_4_). One month after the seedlings were transferred, the Se foliar application was performed for three consecutive weeks (once a week). The Se solution was evenly applied to the upper side of the lettuce leaves with the help of a soft brush in the early morning before sunrise. A wetting agent (0.04% Silwet) was added to the Se solution for better absorption, and each plant received 5 mL of the Se solution. The lettuce plants were harvested on the 53rd day of the experiment. The heads of the lettuce were separated from the roots, and the fresh weight (yield) was recorded. The lettuce heads were washed with distilled water, were frozen in liquid nitrogen, and were stored in a cooling chamber at −20 °C. The DM was determined after the lettuce heads were dried at −53 °C in a freeze dryer (Gamma1-20, Christ, Osterode am Harz, Germany). The dried lettuce heads were ground to a fine powder and were kept for further analyses.

### 4.2. Amino Acids Analysis

For the sample extraction, to determine the amino acids, 100 mg of the plant material was extracted with a chloroform/methanol mixture under an ice bath for one hour. Then, 3 mL of ultrapure water was added to the plant sample, after which the sample was shaken and centrifuged. The aqueous phase was removed, and the sample was shaken a second time with ultrapure water. The two aqueous phases were mixed with 10 μL of octanol and were concentrated using a rotary evaporator. The resulting residue was dissolved in 2 mL of ultrapure water and was stored frozen until further processing. The samples were purified using an RP-18 cartridge. The purified sample was acidified with a sample dilution buffer and was then measured with an amino acid analyzer (Biochrom 30, Biochrom Ltd., Cambridge, UK).

### 4.3. Determination of Inorganic Anions, Organic Acids, and Water-Soluble Sugars

The anions, organic acids, and soluble sugars were determined by using the analytical method of ion chromatography (IC) (IC-5000 Dionex, Thermo Scientific, Waltham, MA, USA). The lyophilized and ground plant samples were prepared for the IC measurement by using hot water extraction. Ultrapure water (1 mL) was added to 20 mg of the sample, and the mixture was shaken thoroughly and placed in a boiling water bath for 5 min. Immediately afterward, the samples were transferred to an ice bath for 30 min. Then, the samples were centrifuged. The supernatant solution of the processed samples was removed and diluted, and a few drops of chloroform were added to it. This led to protein precipitation. The samples were centrifuged a second time, and the resulting supernatants were purified by using a C-18 cartridge with a filter. The purified samples were stored frozen until they were analyzed by using IC.

### 4.4. Mineral Analyses Using ICP-MS

The concentrations of the macronutrients, including magnesium (Mg), phosphorus (P), potassium (K), and calcium (Ca), and of the micronutrients, iron (Fe), manganese (Mn), copper (Cu), zinc (Zn), and Se, were determined by using mass spectrometry with inductively coupled plasma (ICP-MS) (Agilent Technologies 7700 Series, Boebelingen, Germany) in accordance with DINEN ISO17294-2 (Deutsches Institut für Normung, Berlin, Germany, 2005) as described by Abdalla et al. [33].

### 4.5. Determination of the S and N Concentrations Using an Elemental Analyzer

In the first step, to determine the S and the nitrogen (N) concentrations, 6 ± 0.5 mg of each of the lyophilized plant samples were weighed into tin capsules (5 × 9 mm). Then, the tin capsules were folded into cube-shaped packets. For the calibration, five standards were measured at the start, and one standard was measured after every nine plant samples to check the results. The measurement was performed using an elemental analyzer (Flash EA1112, Thermo Fisher Scientific, Milano, Italy).

### 4.6. Statistical Analysis

The data of the growth, Se and S concentrations, elemental composition, amino acids, inorganic anions, organic acids, and water-soluble sugars were statistically analyzed by using a two-way (treatment × cultivar) analysis of variance (ANOVA). The statistical analysis was conducted by using the statistical program, Statistics 10, Version 10.0. A two-factor ANOVA was performed using the software. This was followed by a multiple comparison that was based on the adjustment method using Tukey’s multivariate distribution. The mean values were compared for all of the combinations of the factor levels at a test level of α = 0.05. The software program, GraphPad Prism, Version 8.4.2, was used to create the bar graphs.

## 5. Conclusions

Dramatically, our results provide important information on the effects of varied Se and S concentrations on the plant growth and primary metabolism. Our overall results confirm the substantial effect of Se and S enrichment on butterhead lettuce. The increase in the Se level, particularly, might boost human health, with the added benefits of inducing various dietary minerals. Additionally, the crosstalk between Se and S demonstrated a unique synergistic effect by enhancing the accumulation of Asp and Glu, which induced the synthesis of the amino acids that are related to the Asp and Glu families. Additionally, the glucose, fructose, and sucrose levels accumulated under the Se1/S1 conditions. Moreover, the Se foliar application caused a synergistic relationship with S, which was added to the root via a nutrient solution. This beneficial synergism led to the reduction in the NO_3_^−^ and PO_4_^3−^ concentrations at the higher S level under lower or higher Se treatments. This study encourages further research on the determination and quantification of Se species, including SeMet, SeCys, and Se-Methylselenocysteine (SeMeSeCys), which are metabolized, and accumulate as an organic Se. Because of the chemical similarity between the SeO_4_ and SO_4_ in crop plants, we cannot discriminate between Se and S, as they share common metabolic pathways, including uptake, translocation, and assimilation. The quantification of the Se species will make sense when Se-and-S crosstalk occurs. Hence, selenoamino acids may replace their S-analogues, including Cys and Met, in proteins. This may lead to the impairment of protein folding and function, which may end the overall disruption of the cell metabolism.

## Figures and Tables

**Figure 1 plants-11-00927-f001:**
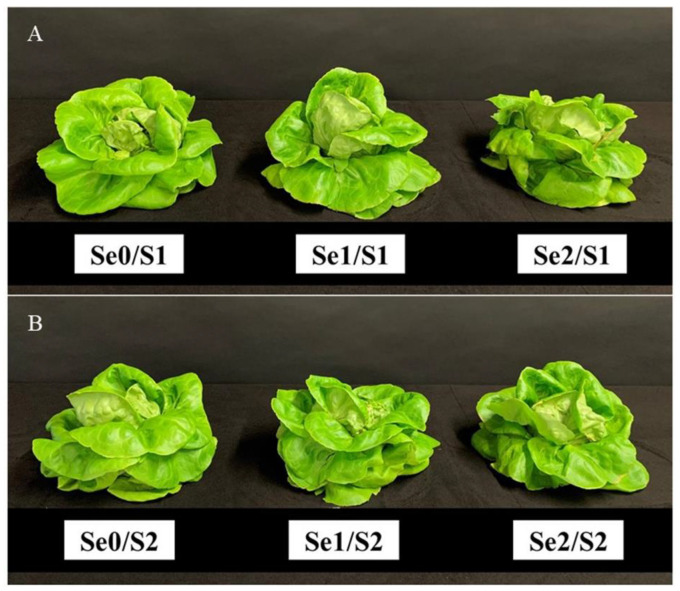
Comparison of the lettuce heads (cultivar: Pazmanea RZ) from a hydroponic system treated with two S levels (S1: 1 mM; and S2: 2 mM (K_2_SO_4_)) and three Se levels (Se0: 0 µM; Se1: 1.3 µM; and Se2: 3.8 µM (Na_2_SeO_4_)). (**A**) Side view of the lettuce heads treated with the S1 level. (**B**) Side view of the lettuce heads treated with the S2 level.

**Figure 2 plants-11-00927-f002:**
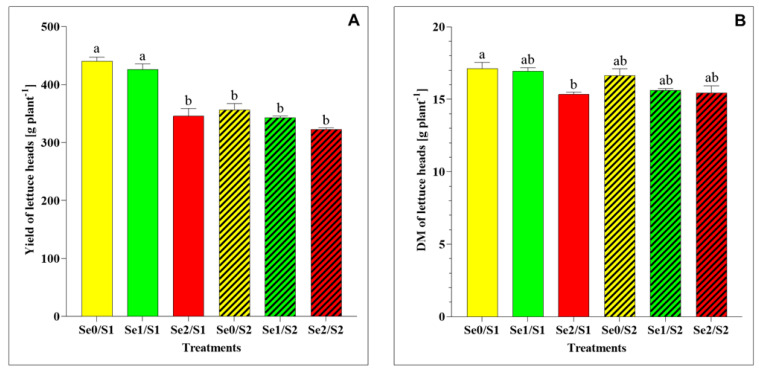
(**A**) Yield and (**B**) dry matter (DM) accumulation in lettuce plants grown in a hydroponic system and treated with two S levels (S1: 1 mM; and S2: 2 mM (K_2_SO_4_)), and three Se levels (Se0: 0 µM; Se1: 1.3 µM; and Se2: 3.8 µM (Na_2_SeO_4_)). The data presented are the means ± standard error of the means (SEMs) of four replicates. Different letters show statistically significant differences among all the treatments (*p* ≤ 0.05; Tukey’s test).

**Figure 3 plants-11-00927-f003:**
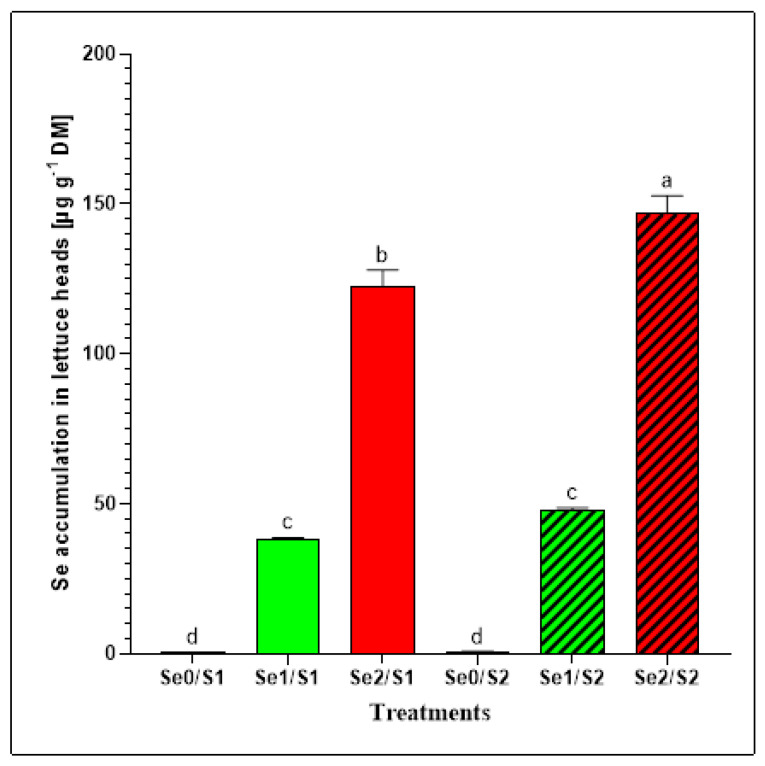
Se accumulation in the lettuce plants grown in a hydroponic system and treated with two S levels (S1: 1 mM; and S2: 2 mM (K_2_SO_4_)), and three Se levels (Se0: 0 µM; Se1: 1.3 µM; and Se2: 3.8 µM (Na_2_SeO_4_)). The data presented are the means ± SE of four replicates. Different letters show statistically significant differences among all the treatments (*p* ≤ 0.05; Tukey’s test).

**Figure 4 plants-11-00927-f004:**
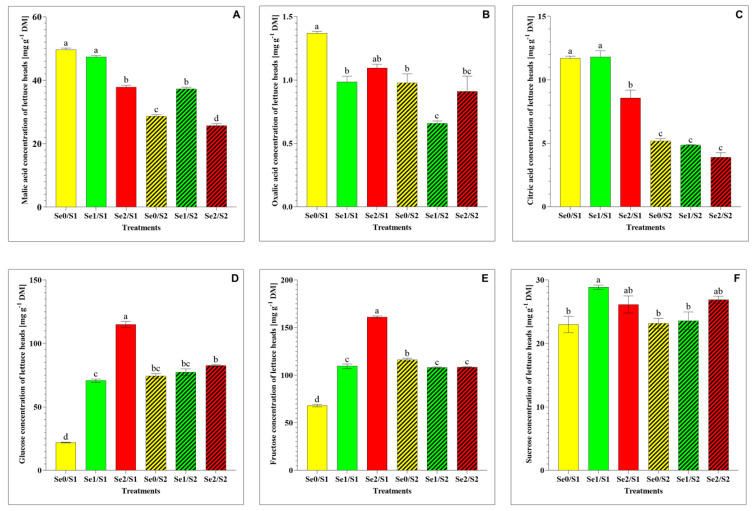
(**A**) malic acid; (**B**) oxalic acid; (**C**) citric acid; (**D**) glucose; (**E**) fructose; and (**F**) sucrose accumulation in the lettuce plants grown in a hydroponic system and treated with two S levels (S1: 1 mM; and S2: 2 mM (K_2_SO_4_)), and three Se levels (Se0: 0 µM; Se1: 1.3 µM; and Se2: 3.8 µM (Na_2_SeO_4_)). The data presented are the means ± SE of four replicates. Different letters show statistically significant differences among all the treatments (*p* ≤ 0.05; Tukey’s test).

**Table 1 plants-11-00927-t001:** Macronutrients (mg g^−1^ DM) and micronutrients (μg g^−1^ DM) concentrations in the lettuce plants grown in a hydroponic system and treated with two S levels (S1: 1 mM; and S2: 2 mM (K_2_SO_4_)), and three Se levels (Se0: 0 µM; Se1: 1.3 µM; and Se2: 3.8 µM (Na_2_SeO_4_)).

Treatments						
	Se0/S1	Se1/S1	Se2/S1	Se0/S2	Se1/S2	Se2/S2
**Macronutrients**						
P	3.81 ± 0.03 ^c^	4.10 ± 0.08 ^ab^	4.22 ± 0.1 ^a^	3.81 ± 0.04 ^c^	4.06 ± 0.1 ^b^	4.21 ± 0.1 ^ab^
Mg	1.28 ± 0.1 ^b^	1.32 ± 0.1 ^b^	1.27 ± 0.02 ^b^	1.50 ± 0.1 ^a^	1.52 ± 0.03 ^a^	1.60 ± 0.04 ^a^
K	49.9 ± 0.7 ^a^	50.3 ± 0.2 ^a^	46.1 ± 0.8 ^b^	43.2 ± 0.7 ^c^	43.6 ± 0.4 ^c^	44.1 ± 0.9 ^c^
Ca	2.4 ± 0.1 ^b^	2.62 ± 0.1 ^a^	2.21 ± 0.1 ^c^	2.00 ± 0.1 ^d^	2.00 ± 0.03 ^d^	2.1 ± 0.1 ^d^
N	37.7 ± 0.87 ^a^	36.6 ± 0.6 ^ab^	36.7 ± 0.9 ^ab^	36.01 ± 0.5 ^bc^	34.9 ± 0.6 ^c^	34.7 ± 0.9 ^c^
S	2.97 ± 0.04 ^bc^	3.09 ± 0.1 ^ab^	2.85 ± 0.1 ^c^	2.91 ± 0.1 ^c^	3.09 ± 0.1 ^ab^	3.13 ± 0.1 ^a^
**Micronutrients**						
Mn	24.00 ± 0.9 ^c^	29.2 ± 0.7 ^b^	31.3 ± 1.6 ^a^	13.9 ± 0.3 ^d^	13.4 ± 0.4 ^d^	13.9 ± 0.23 ^d^
Fe	37.3 ± 0.8 ^d^	37.6 ± 1.2 ^d^	38.7 ± 1.0 ^d^	41.1 ± 0.8 ^c^	46.6 ± 0.5 ^a^	43.3 ± 0.6 ^b^
Cu	5.57 ± 0.1 ^b^	6.07 ± 0.03 ^a^	4.94 ± 0.12 ^c^	4.83 ± 0,22 ^c^	5.02 ± 0.06 ^c^	2.94 ± 0.03 ^d^
Zn	91.4 ± 0.5 ^b^	98.00 ± 1.8 ^a^	53.00 ± 1.3 ^d^	97.2 ± 0.2 ^a^	90.3 ± 0.6 ^b^	59.2 ± 1.00 ^c^

The data presented are the means ± SE of four replicates. Different letters in the rows show statistically significant differences among all the treatments (*p* ≤ 0.05; Tukey’s test).

**Table 2 plants-11-00927-t002:** Amino acids accumulation in lettuce plants grown in a hydroponic system and treated with two S levels (S1: 1 mM; and S2: 2 mM (K_2_SO_4_)), and three Se levels (Se0: 0 µM; Se1: 1.3 µM; and Se2: 3.8 µM (Na_2_SeO_4_)).

Treatments						
Amino Acids	Se0/S1	Se1/S1	Se2/S1	Se0/S2	Se1/S2	Se2/S2
**Asp**	3.86 ± 0.1 ^c^	4.69 ± 0.2 ^a^	4.59 ± 0.2 ^a^	3.71 ± 0.2 ^c^	4.22 ± 0.3 ^b^	4.67 ± 0.02 ^a^
**Thr**	9.41 ± 0.1 ^a^	9.28 ± 0.3 ^a^	7.68 ± 0.2 ^c^	7.02 ± 0.3 ^d^	8.36 ± 0.4 ^b^	8.87 ± 0.04 ^ab^
**Ser**	16.60 ± 0.7 ^a^	16.98 ± 0.7 ^a^	14.46 ± 0.4 ^b^	13.6 ± 0.6 ^b^	16.5 ± 0.9 ^a^	17.1 ± 0.1 ^a^
**Asn, Glu, Gln**	41.00 ± 2.4 ^bc^	45.4 ± 1.9 ^ab^	39.00 ± 2.2 ^cd^	35.5 ± 1.9 ^d^	43.9 ± 2.7 ^b^	48.7 ± 0.3 ^a^
**Gly**	1.21 ± 0.1 ^ab^	1.12 ± 0.1 ^bc^	1.47 ± 0.2 ^a^	0.86 ± 0.1 ^c^	1.16 ± 0.1 ^b^	1.04 ± 0.1 ^bc^
**Ala**	17.2 ± 0.3 ^a^	15.4 ± 0.8 ^b^	15.3 ± 1.2 ^b^	11.3 ± 0.8 ^d^	13.6 ± 0.7 ^c^	14.00 ± 0.9 ^bc^
**Val**	12.2 ± 0.25 ^a^	11.9 ± 1.00 ^a^	10.5 ± 1.1 ^b^	9.9 ± 0.5 ^b^	11.9 ± 0.3 ^a^	12.6 ± 0.3 ^a^
**Ile**	4.17 ± 0.1 ^a^	4.07 ± 0.2 ^a^	3.74 ± 0.1 ^b^	3.06 ± 0.2 ^c^	4.10 ± 0.1 ^a^	4.08 ± 0.03 ^a^
**Leu**	2.57 ± 0.3 ^a^	2.22 ± 0.2 ^a^	2.41 ± 0.1 ^a^	1.46 ± 0.1 ^b^	2.49 ± 0.1 ^a^	2.11 ± 0.1 ^a^

The data presented are the means ± SE of four replicates. Different letters in the rows show statistically significant differences among all the treatments (*p* ≤ 0.05; Tukey’s test).

**Table 3 plants-11-00927-t003:** Inorganic anions concentrations in the lettuce plants grown in a hydroponic system and treated with two S levels (S1: 1 mM; and S2: 2 mM (K_2_SO_4_)), and three Se levels (Se0: 0 µM; Se1: 1.3 µM; and Se2: 3.8 µM (Na_2_SeO_4_)).

Treatments						
	Se0/S1	Se1/S1	Se2/S1	Se0/S2	Se1/S2	Se2/S2
**Cl^−^**	22.8 ± 1.9 ^a^	18.7 ± 0.6 ^a^	15.7 ± 0.8 ^b^	11.5 ± 1.1 ^c^	8.95 ± 0.6 ^d^	9.16 ± 0.86 ^cd^
**SO_4_^2−^**	1.50 ± 0.2 ^a^	1.09 ± 0.04 ^b^	0.69 ± 0.1 ^c^	0.96 ± 0.11 ^bc^	0.81 ± 0.04 ^cd^	0.80 ± 0.1 ^cd^
**NO_3_^−^**	14.9 ± 0.6 ^a^	11.3 ± 0.4 ^b^	8.94 ± 0.6 ^c^	9.04 ± 0.58 ^c^	6.72 ± 0.51 ^d^	6.85 ± 0.83 ^d^
**PO_4_^3−^**	5.68 ± 0.9 ^a^	4.68 ± 0.7 ^ab^	4.34 ± 1.01 ^bc^	3.26 ± 0.14 ^c^	3.37 ± 0.38 ^c^	3.20 ± 0.45 ^c^

The data presented are the means ± SE of four replicates. Different letters in the rows show statistically significant differences among all the treatments (*p* ≤ 0.05; Tukey’s test).

## Data Availability

Data are provided in the article.

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
