# Peer review of "Crosstalk between Selenium and Sulfur Is Associated with Changes in Primary Metabolism in Lettuce Plants Grown under Se and S Enrichment"

_plants, 2022, doi:10.3390/plants11070927_

Round 1

Reviewer 1 Report

The research topic of this paper is interesting and the results of the Se and S crosstalk could be very useful in determination of the nutritional value of the lettuce. Manuscript is well written; background, methods and discussion are very descriptive and easily followed; cited references are recent and applicable. However, my greatest concern in this research is the sample size. Four plants in each group is pretty small number, and great conclusions (“dramatically decrease” etc) are rather pretentious for such a small sample size. Also, SD is unusually low in most of the analysis. Furthermore, is it appropriate to determine the SO42- level in plants treated with different concentrations of K2SO4?

Author Response

Reviewer 1.

Comments and Suggestions for Authors

The research topic of this paper is interesting and the results of the Se and S crosstalk could be very useful in determination of the nutritional value of the lettuce. Manuscript is well written; background, methods and discussion are very descriptive and easily followed; cited references are recent and applicable.

However, my greatest concern in this research is the sample size. Four plants in each group is pretty small number, and great conclusions (“dramatically decrease” etc) are rather pretentious for such a small sample size.

Response:

We are grateful to the reviewer for the insightful comments on our manuscript. With respect to the reviewer´s comment, we believe that our experiment is sufficient since all the parameters of good and quality research have adhered. Additionally, we think that the study as being exploratory and hypothesis-generating exhibited more a qualitative than quantitative analysis, hence we do not oversell the findings. Moreover, we did a preliminary experiment on a small scale and it showed the same results. Furthermore, we still have unpublished results, which agree with the current findings. The word (“dramatically”) has changed to significantly (Line 350). A correction was highlighted in yellow colour.

Also, SD is unusually low in most of the analysis.

Response:

Thank you very much for the reviewer’s valuable comment. We are sorry for this mistake, as actually, we used the standard error of the mean (SEM or SE) for all figures and tables and not standard deviation (SD). Corrections have been performed as highlighted in the manuscript file (Lines 116, 136, 166, 190, 226, 232).

Furthermore, is it appropriate to determine the SO42- level in plants treated with different concentrations of K2SO4?

Response:

Thank you for the reviewer’s comment. With respect to the reviewer´s comment, we did determine SO42- (Table 3) for all lettuce plant samples. The study found that SO42− declined significantly in response to Se1 and Se2 under the S1 conditions (by 27% and 37%, respectively), while the SO42− concentration remained unaffected under the higher S (S2) and two varied Se conditions (Line 212214). The corrections have been performed as highlighted in the manuscript file.

Reviewer 2 Report

The article entitled “Selenium and sulfur crosstalk is associated with changes in primary metabolism in lettuce plants grown under Se and S enrichment" presents the results of a study which aimed to determine the effects of Se and S supply on the biosynthesis of amino acids, water-soluble sugars, and organic acids in butterhead lettuce.

General note

The subject of the study is very interesting and topical, with scientific and practical importance.

The introduction is presented correctly, in accordance with the subject Numerous scientific articles, in concordance to the topic of the study, were consulted.

Methodology of the study was clearly presented, and appropriate to the proposed objectives.

The obtained results are important and have been analyzed and interpreted correctly, in accordance with the current methodology.

The discussions are appropriate, in the context of the results, and was conducted compared to other studies in the field.

The scientific literature, to which the reporting was made, is recent and representative in the field.

There are some minor changes I am suggesting in detailed comments below.

The description of the methodology lacks information on the number of repetitions of experimental treatments. Please add.

Below Table 1 is a description: Different letters show statistically significant differences among all the treatments. Does this mean that the analysis was not performed separately for individual macro and micronutrients? If the analysis was conducted separately "Different letters in the rows ..." should be added.

A similar note to the description in tables 2 and 3.

Author Response

Reviewer 2.

Comments and Suggestions for Authors

The article entitled “Selenium and sulfur crosstalk is associated with changes in primary metabolism in lettuce plants grown under Se and S enrichment" presents the results of a study that aimed to determine the effects of Se and S supply on the biosynthesis of amino acids, water-soluble sugars, and organic acids in butterhead lettuce.

General note

The subject of the study is very interesting and topical, with scientific and practical importance.

The introduction is presented correctly, in accordance with the subject Numerous scientific articles, in concordance to the topic of the study, were consulted.

Methodology of the study was clearly presented, and appropriate to the proposed objectives.

The obtained results are important and have been analyzed and interpreted correctly, in accordance with the current methodology.

The discussions are appropriate, in the context of the results, and was conducted compared to other studies in the field.

The scientific literature, to which the reporting was made, is recent and representative in the field.

There are some minor changes I am suggesting in detailed comments below:

The description of the methodology lacks information on the number of repetitions of experimental treatments. Please add.

Response:

Thank you for the reviewer’s positive comment. The number of repetitions of experimental treatments was included; “There were four replicates arranged in a completely randomized design” (Lines 382–383). All corrections were highlighted in yellow colour.

Below Table 1 is a description: Different letters show statistically significant differences among all the treatments. Does this mean that the analysis was not performed separately for individual macro and micronutrients? If the analysis was conducted separately "Different letters in the rows ..." should be added.

A similar note to the description in tables 2 and 3.

Response:

Thank you for the reviewer’s comment. Changes were performed in line with the reviewer’s comment. “Different letters in the rows” were added to the description in tables 1, 2, and 3 (Lines 166, 190, and 232). All corrections were highlighted in yellow colour.

Reviewer 3 Report

In this paper, Abdalla et al. examined Selenium and sulfur crosstalk is associated with changes in pri-2 mary metabolism in lettuce plants grown under Se and S enichment. They showed important information on the effects of varied Se and S concen-457 trations and their interaction on plant growth and primary metabolism. Our overall re-458 sults confirmed the substantial effect of Se and S enrichment on butterhead lettuce. They indicated that the Se foliar application caused a synergistic relationship with S, which was added to the 465 root via nutrient solution..

I have read at great length the work. I found this article informative in regards to background information. The manuscript appears to have sufficient scientific quality and may be of interest to the readers of Plants.

The article is sufficiently novel and interesting to warrant publication

The article is clearly laid out.

Abstract reflects the content of the article.

Introduction is sufficient.

Methods are suitable to reach the results.

The author(s) explain clearly laid out and in a logical sequence what he/she/they discovered in the research.

Discussion is good.

References are enough.

Author Response

Response:

Thank you for the reviewer’s positive comment.

Reviewer 4 Report

I find this article well written and I recommend to publish it in present form.

Author Response

(The authors gave the same response as above.)

Round 2

Reviewer 1 Report

The resubmitted article now includes all corrections as responses to my comments and it can be accepted for publication in present form.

Best regards